



# Modelling of the public health costs of fine particulate matter and results for Finland in 2015

Kukkonen, Jaakko[1], Mikko Savolahti[2], Yuliia Palamarchuk[1], Timo Lanki[3], Väinö Nurmi[1], Ville-Veikko Paunu[2], Leena Kangas[1], Mikhail Sofiev[1], Ari Karppinen[1], Androniki Maragkidou[1], Pekka Tiittanen[3], Niko Karvosenoja[2]

[1] Finnish Meteorological Institute, Erik Palmenin aukio 1, P.O. Box 503, 00101, Helsinki, Finland
[2] Finnish Environment Institute, Latokartanonkaari 11, FI-00790 Helsinki, Finland
[3] National institute for health and Welfare, PL 30, 00271 Helsinki, Finland

Correspondence: Jaakko Kukkonen (jaakko.kukkonen@fmi.fi)

**Abstract.** We have developed an integrated tool of assessment that can be used for evaluating the
public health costs caused by the concentrations of fine particulate matter ($PM_{2.5}$) in ambient air. The model can be used in assessing the impacts of various alternative air quality abatement measures, policies and strategies. The model has been applied for the evaluation of the costs of the domestic emissions that influence the concentrations of $PM_{2.5}$ in Finland in 2015. The model includes the impacts on human health; however, it does not address the impacts on climate change or the state of the
environment. First, the national Finnish emissions were evaluated using the Finnish Regional Emission Scenarios model (FRES) on a resolution of 250 x 250 $m^2$ for the whole of Finland. Second, the atmospheric dispersion was analyzed by using the chemical transport model SILAM and the source-receptor matrices contained in the FRES model. Third, the health impacts were assessed by combining the spatially resolved concentration and population datasets, and by analyzing the impacts for various
health outcomes. Fourth, the economic impacts for the health outcomes were evaluated. The model can be used to evaluate the costs of the health damages for various emission source categories, for a unit of emissions of $PM_{2.5}$. It was found that economically the most effective measures would be the reduction of the emissions in urban areas of (i) road transport, (ii) non-road vehicles and machinery, and (iii) residential wood combustion. The reduction of the precursor emissions of $PM_{2.5}$ was clearly less
effective, compared with reducing directly the emissions of $PM_{2.5}$. We have also designed a user-friendly web-based tool of assessment that is available open access.

## 1. Introduction

Air pollution related to particulate matter (PM) can result in a wide variety of impacts. Prominent examples of these include enhancement or mitigation of climate change, adverse impacts on the health of the populations, and various consequences on the environment (e.g., influence on biodiversity, acidification and eutrophication). Air pollution may also cause corrosion of materials and degradation
of buildings and cultural heritage (e.g., Al-Thani et al., 2018). This study focuses on the impacts of air pollution on public health. The projected economic growth, urbanization and the increased fraction of senior population will increase the effects on public health in some regions in the future (e.g., OECD, 2016).





Emission standards and other control policies in many cases address only the amounts of emissions. Such policies will not be optimal for the mitigation of the impacts of poor air quality, as the same amount of emissions from different sources may have totally different damage costs (e.g., Muller and Mendehlson, 2009; Carson and LaRiviere, 2018). Economists have therefore suggested market-based approaches, such as emission taxes (e.g., Baumol and Oates, 1998) or tradable permits. As the marginal

damages (defined as the additional damage caused by an additional unit of emission) and the properties of the emission sources, such as emission heights, differ across regions (Nahlik et al., 2016), environmental policies should reflect these differences. It is therefore worthwhile to evaluate the relative costs for potential emission reductions from different emission source categories located in different regions.


There is a fairly extensive amount of scientific literature regarding the cost evaluations of air pollution on public health, including especially the effects of the $PM_{2.5}$ concentrations. Muller and Mendehlson (2009), Holland et al. (2015) and Heo et al. (2016) have evaluated the unit costs for the emissions at various stack heights on a fine spatial resolution in the United States at a county level. Buoconore et al.

(2014), Levy et al. (2009) and Fann et al. (2009) have conducted similar studies on a coarser resolution in the United States. Moreover, Nahlik et al. (2016) estimated the county-specific unit damage costs for PM (especially $PM_{2.5}$), in addition to SOx, NOx and VOCs at major airports in the United States. Trejo-González et al. (2019) analyzed economic costs associated with exposure to $PM_{2.5}$ in 2013 and 2015 in Mexican cities assuming two mitigation scenarios. In Europe, Holland et al. (2014) and Brandt

et al. (2010) have evaluated unit costs at country level. Defra (2015) and Walton et al. (2015) have conducted similar studies regionally in Europe. In Asia, and more specifically in China, Qi et al. (2018) investigated the losses as well as the consequences caused in the economy by ambient $PM_{2.5.}$

The study by OECD (2016a) pointed out that impacts due to $PM_{2.5}$ concentrations commonly

contribute to more than 90 % of the total health costs of air pollution. Clearly, the exact proportion of these effects substantially depends on the domain and the year of evaluation. Emissions of the most important $PM_{2.5}$ precursors, such as $NO_x$, $SO_2$ and $NH_3$, have also been included in some studies (e.g., Walton et al., 2015). The direct health costs of $NO_2$ and $O_3$ may also be substantial in some cases.

With respect to unit cost modelling, most studies have used the so-called impact pathway approach. This approach combines air quality modelling with population data, epidemiological evidence and economic modelling (Im et al., 2018). It is a sequential approach, in which one assumes a change in emissions, models the corresponding changes in air quality, uses epidemiological evidence to calculate the health response, and finally applies economic evidence. For instance, Trejo-González et al. (2019)

concluded in their study that a reduction of the annual $PM_{2.5}$ average to less than 10 μg/m$^3$ in 2015 would have decreased mortality by 14,666 (avoidable deaths) with estimated costs of 64,164 million dollars in Mexican cities.

Some previous studies have used chemical transport models on regional or continental scales (e.g.,

Fann et al., 2009; Buonore et al., 2014; Brandt et al., 2010; Im et al., 2018). Another approach is to use simplified decision-support modelling systems that use pre-computed atmospheric dispersion statistics or source - dispersion matrices (e.g., Muller and Mendehlson, 2009, Holland et al., 2014, 2015, Bickel et al., 2003). One example of such approaches was presented by Heo et al. (2016); they attempted to generalize the results of chemical transport models using statistical methods. As this approach

substantially reduces the computational effort, one can evaluate a much larger number of various





emission reduction scenarios. Heo et al. (2016) computed the resulting changes in air quality for a one ton reduction in emissions for 11 different emission sources in United States.

In the next stage of the evaluation, one will evaluate the health impacts caused by the changes in the concentrations. Some of the studies have included only the increased risk of early mortality. (e.g. Heo et al. 2016; Buonocore et al. 2014; Levy et al. 2009), due to the fact that mortality costs commonly dominate the total unit costs. In these studies, $PM_{2.5}$ induced mortality has been modelled with a linear response-function model, in which an increase in the concentration levels is linearly translated into either to loss of human lives or years of life lost (YOLL) years. For example, 144,289 and 150,771 potential YOLLs due to exposure to $PM_{2.5}$ were estimated for 2013 and 2015, respectively, in Mexican cities (Trejo-González et al. (2019) The response functions have been estimated in epidemiological studies, such as Pope et al. (2002) or based on a combination of other studies related to long-term exposure to $PM_{2.5}$ and $PM_{10}$, such as Trejo-González et al. (2019). However, most studies have also included other end-points; commonly at least morbidity costs (Muller and Menhdelson, 2009; Holland et al. 2016; Fann et al. 2009; Walton et al. 2015; Defra, 2015; EEA, 2014). Some studies (Muller and Mendehlson, 2009; Walton et al. 2015) have also included the loss of agricultural yields; however, resulting on minor effect on the unit costs.In a more recent study conducted by Trejo-González et al. (2019) the lost productivity was also calculated for 2013 and 2015 in Mexican cities for different age groups (15 and more, 30 and more, and 25 to 74 years). In China, Qi et al. (2018) estimated that the total national loss due to exposure to $PM_{2.5}$ was 79.2 billion RMB ¥.

As the increased risk of early mortality commonly dominate the unit cost estimates, the assumptions behind its computation explain a large fraction of the variation in various damage cost estimates. The health response-functions contain a risk ratio or relative risk (RR) for an increase in concentration of 10 $\mu g/m^3$ that describes the change in the relative risk level. Relative risk (RR) is generally defined as the ratio of the probability of an outcome in an exposed group to the probability of an outcome in an unexposed group. Moreover, RR is different from one region to another depending on ambient $PM_{2.5}$ composition and the variation in people's sensitivity (Qi et., 2018) A low value was applied by Bicket et al. (2003), RR = 1.024, whereas Pope et al. (2002) estimated a much higher value, RR = 1.077. The latter estimate has been widely used in unit cost studies (Muller and Mendehlson, 2009; Holland et al. 2016; EEA, 2014). Qi et al. (2018) also applied a low RR for lung cancer related to $PM_{2.5}$ in China and it was equal to 1.03. The same value was used by Cao et al. (2011) and Loomis, Huang, and Che (2014). American Cancer Society published an estimate of 1.075 that was used in Heo et al. (2016). A more conservative estimate of 1.06 has been reported in some studies such as Defra et al. (2015 and Raza et al. (2018), apart from Woodcock et al. (2009; 2013;2014) and Dhondt et al. (2013). The Harvard Six Cities -study (Laden et al., 2006) resulted in an even more substantial mortality, i.e., RR = 1.12. This value has also been used widely (Fann et al. 2009; Levy et al. 2009). Raza et al. (2018) presented an even higher RR for $PM_{2.5}$ (RR=1.17) in their paper which was originally reported in another study regarding air pollution and mortality in Los Angeles (Jerrett et al., 2005)

Next step in the analysis chain is to convert the health impacts into monetary values. With respect to mortality, there are two main approaches for the monetary valuation: either (i) counting the expected value of life years lost and multiplying with the value of a life year (VOLY), or (ii) counting the expected value of early mortality and multiplying with the value of life (VSL). However, both the values of VOLY's and those of VSL's, and the final cost results obtained using these two approaches can vary substantially. Regarding the VSL, a fairly low estimated value in Muller and Mendehlson (2009) was two million dollars, with an age-adjusted value of 1.2 million dollars, whereas Heo et al.



(2016) evaluated VSL to be 8.6 million dollars. VSL was equal to 1.629 and 1.643 million dollars in 2013 and 2015, respectively, in Mexican cities of the National Urban System (Trejo-González et al. 2019) EU-based studies have commonly indicated a higher public health cost value using the VSL-method, compared with those obtained using VOLY; e.g., the study by EEA (2014) found that the VSL-based values were approximately 2.5 higher than the VOLY-based values.

Taking into account the concentrations nowadays and during the past decade, particulate matter can be considered in most locations to be more harmful than gaseous pollutants; e.g., this has been found to be the case for the Nordic countries by Hänninen et al. (2016), Lehtomäki et al. (2018) and Kukkonen et al. (2018). WHO (2013a) has evidenced a strong association between the concentrations of coarse and ultrafine particles and harmful effects. However, the majority of epidemiological studies have focused on $PM_{2.5}$, or alternatively on $PM_{10}$, including $PM_{2.5}$ as a sub-fraction, and therefore the most established
concentration-response functions have been developed for these size fractions. This study therefore primarily focuses on fine particulate matter.

The overarching aim of this study is to develop an integrated tool of assessment for evaluating the public health costs caused by the ambient air concentrations of fine particulate matter ($PM_{2.5}$). The
objectives of this study were (i) to present an impact pathway model for evaluating the public health costs due to the concentrations of $PM_{2.5}$, (ii) to present selected example results regarding the various stages of this assessment for domestic pollution sources in Finland in 2015, and (iii) to present both an easy-to-use summary tabulation and a web-based computation system of the public health costs for various emission categories. The final model framework includes emission and dispersion modelling,
health impact assessment and economic evaluation. The final model and results regarding the costs of the emissions from various source categories can be used to assess the impacts of national and urban scale air quality strategies as well as to compare the cost-efficiency of various potential emission mitigation measures. The model framework could be also adapted for similar economic cost analyses in other countries or geographical domains in the future.


## 2  Methods

This study adopts the impact pathway approach, to combine the various modelling stages.

### 2.1 Inventory of the domestic emissions

The anthropogenic emissions in Finland in 2015 were computed using the Finnish Regional Emission
Scenarios model (FRES). For a detailed description of the FRES model, the reader is referred to Karvosenoja (2008), Karvosenoja et al. (2011) and Savolahti et al. (2016). The modelling included the anthropogenic emissions of the compounds $PM_{10}$, $PM_{2.5}$, $PM_1$, BC (black carbon), OC (organic carbon), mineral dust, $SO_2$, $NO_x$, $NH_3$, NMVOC and CO. The emissions were computed on a grid of 250 m x 250 m for the whole of Finland, for various area sources. In addition, the modelling included
424 industrial point sources. For the latter, coordinates and stack heights were used that were specific for each installation (Karvosenoja et al., 2011).





The emission scenarios included the most significant pollutants for each source category. These included the primary emissions as follows: $PM_{2.5}$, $NO_x$ and $SO_2$ for industrial installations and power
plants, $PM_{2.5}$ and $NO_x$ for vehicular traffic and machineries, $PM_{2.5}$ for residential wood combustion, and $NH_3$ for agriculture. First, we computed a baseline emission scenario for a selected recent year, 2015. Second, the emissions from each of the considered emission sectors and considered pollutants were reduced by a constant moderate percentage, selected to be 10 %, compared with the baseline scenario.


The health damage caused by the population exposure is substantially dependent on the spatial correlation of the distributions of the population and the emission sources (e.g., Soares et al., 2014). Such a correlation can be especially high for vehicular traffic and residential wood combustion. These two emission source categories were therefore separately analyzed for two classes, viz. emissions in
urban and non-urban areas. In this study, urban areas were defined according to two criteria: (i) these had to include grid cells (250 m x 250 m) that contained at least 200 residents, and (ii) buildings had not be further from each other than 200 m.

For point sources, we have also treated the $PM_{2.5}$ emissions separately, depending on the location of the
facility. This was done, as the population density in the vicinity of various locations varied substantially. We have therefore evaluated separately the unit costs for (i) the Helsinki capital area, (ii) the municipalities of more than 50 000 inhabitants and (iii) the other areas.

**2.2 Atmospheric dispersion modelling**

We have evaluated the atmospheric dispersion using two models: (i) the chemical transport model SILAM (e.g., Sofiev et al, 2006 and 2015), and (ii) the source receptor matrices contained in the FRES model (Karvosenoja et al., 2011). The SILAM model can be used for regional, continental and global
scale evaluations, whereas the FRES model is applicable on local and regional scales.

We have used two models, as both their applicability and results are complementary. The model computations using the SILAM model include also the long-range transported contributions from the rest of Europe, whereas the FRES computations address only the dispersion of the domestic emissions.
Another advantage of the SILAM model computations is that the formation of secondary $PM_{2.5}$ is taken into account, whereas these are not included in the FRES model computations. On the other hand, the FRES computations are substantially less resource-consuming, and we therefore could execute the model on a very fine spatial resolution, 250 x 250 $m^2$. In this study, we used the SILAM computations on a resolution of 5 x 5 $km^2$ over the Finnish domain.


The impacts of the various domestic emission reduction scenarios were evaluated by numerically changing the Finnish emissions of a selected source category, whereas the emissions from the other domestic source categories were kept the same. In the SILAM computations, the emissions from the rest of Europe were also assumed to be the same, for all the emission scenarios. In this way, one can
evaluate the impact of one selected national source category on the concentrations of $PM_{2.5}$.

First, we computed atmospheric dispersion for the baseline emission scenario in 2015, using actual meteorological data for that year. Second, the atmospheric dispersion was computed for the reduced





emission scenarios described above. Finally, the differences of these two computations were computed, and the results were converted to correspond to a reduction of a unit mass of emissions.

### 2.2.1 Modelling using the SILAM model on the European and national scales

SILAM is a dispersion model from global to mesoscales that has been developed for evaluating atmospheric composition. The model is also used for policy guidance in case of emergencies and for solving inverse dispersion problems. The model includes dispersion and transport treatments using both Eulerian and Lagrangian approaches. The model contains eight chemical and physical transformation modules, viz. basic acid chemistry and secondary aerosol formation, ozone formation and

transformation in the troposphere and the stratosphere, radioactive decay, aerosol dynamics and transformation of pollen. The model also includes modules for three- and four-dimensional variational data assimilation (http://silam.fmi.fi/). For a more detailed description of the model, the reader is referred to Sofiev et al. (2015).

The computations using the SILAM model included both global and European scale transport and the contributions from the domestic (Finnish) emission sources. The modelling for the whole of Finland was carried out on a resolution of five kilometers. A detailed description of these computations has been previously presented by Lehtomäki et al. (2018).

The SILAM model computations included also the impacts of the chemical and physical transformations on the formation of secondary $PM_{2.5}$. These reactions include especially the impacts of the emissions of sulphur, nitrogen and ammonia compounds of both natural and anthropogenic origin on the concentrations of $PM_{2.5}$. In the model calculations, the full spectra of emitted compounds were included, taking separately into account the temporal variations for each individual sector. The

modelling also allowed to treat simultaneously the sectoral specifications of the point and area sources. This enabled to estimate independently the contributions of the emission reductions on $PM_{2.5}$ concentrations, originated from power plants, industry, traffic and agricultural ammonia.

### 2.2.2 Modelling using the FRES model on the national scale


The FRES model was applied for the evaluation of the impacts of primary domestic emissions. These computations had a spatial resolution of 250 x 250 $m^2$ over the whole of Finland. The source-receptor matrices that were used in this model were based on the computations using the dispersion model

UDM-FMI (Urban Dispersion Modelling system by the Finnish Meteorological Institute; e.g., Karppinen et al., 2000a).

The UDM-FMI model is based on Gaussian plume equations for multiple sources, including stationary point, area and volume sources. The modelling system including the UDM-FMI model has been

previously extensively evaluated versus urban measurement data for gaseous pollutants (e.g., Karppinen et al., 2000b and Kousa et al., 2001) and for $PM_{2.5}$ (e.g., Kauhaniemi et al., 2008, Kukkonen et al., 2018).

The source receptor matrices were based on separate computations over ten climatic sub-zones in

Finland, assuming two different emission heights. Such computations were necessary, as the dispersion





processes are strongly dependent on the climatic variation of the relevant meteorological conditions. The computations were performed on an hourly basis for a period of five or six years for each of the ten climatic zones, depending on the availability of the relevant meteorological data. In the final computations using the FRES model, monthly average source-receptor matrices were used.


### 2.3 Health impact assessment

In this assessment, we have not explicitly allowed for the health effects caused by the $NO_2$ concentrations. The main reason for this choice is that we have allowed for the health impacts

associated with the secondary $PM_{2.5}$ concentrations that have resulted from the NO and $NO_2$ precursor emissions. Including also the health impacts in case of the $NO_2$ concentrations would therefore probably result in double counting. Another reason for not explicitly including the health impacts of $NO_2$ exposure is that the concentration-response function for $NO_2$ has an effective range for annual average concentrations exceeding 20 ug/m$^3$; the concentrations of $NO_2$ are commonly lower than this

threshold value in the present study.

We have combined the modelled annually averaged concentrations of $PM_{2.5}$ with the population count data provided by Statistics Finland in 2015. These datasets were combined in a 250 x 250 m$^2$ grid, for five-yearly age categories. The health effects of $PM_{2.5}$ were assumed to be linear in the concentration range observed in Finland. It was therefore possible to use annual concentration data for the

computations of the health impacts regarding both short- and long-term exposures.

We have computed the health impacts for each grid cell (i) within the domain (i.e., the whole of Finland). The exposure of the population to the concentrations of $PM_{2.5}$ in a grid cell is

$PE_i = P_i \times C_i$,

where $P_i$ and $C_i$ are the population and concentration in the grid cell i, respectively. For each health outcome, the effect of the $PM_{2.5}$ exposure was estimated by calculating the relative excess risk:

$RER = (RR-1) \times 0.1$,

where RR is the risk ratio for $PM_{2.5}$ for the considered health outcome.

The computation takes into account that risk ratios for $PM_{2.5}$ are usually presented in terms of a 10

µg/m$^3$ increase in concentration. However, for some health outcomes, reliable risk ratios have only been established for $PM_{10}$. In such cases, the RER of $PM_{10}$ multiplied by 1.54 was used, as recommended by WHO (WHO, 2013b). The underlying assumptions in deriving this numerical value were that the $PM_{2.5}$ concentration constitutes 65 % of the $PM_{10}$ concentration, and the health effects of $PM_{10}$ can be explained by $PM_{2.5}$.


The number of cases of a considered health outcome in each grid cell was calculated as follows:

$N_i = PE_i \times RER \times BR$ ,





where BR is the background risk of a considered health outcome. The total impact of PM$_{2.5}$ exposure
      on an outcome was calculated by summing the numbers of cases over all grid cells. We computed the
      total number of years of life lost due to the PM$_{2.5}$ exposure by (i) multiplying the evaluated deaths with
      life-expectancy in one-yearly age categories, and (ii) subsequently summing the lost life years over all
      the age categories.


      The exposure to fine particulate matter has been reported to be associated with a substantial number of
      health outcomes in epidemiological studies (Qi et al., 2018; Raza et al., 2018; Im et al., 2018);
      however, reliable estimates of the concentration-response functions have been derived only for a
      limited number of outcomes. In this study, the functions recommended within the HRAPIE project
were used (WHO, 2013b). These functions have been considered sufficient to enable the quantification
      of both the effects of the long-term PM$_{2.5}$ exposures on mortality, and the short-term exposures on
      cardiovascular and respiratory hospital admissions (Im et al., 2018).

      We did not use any threshold for the PM$_{2.5}$ effects, as even relatively low levels of PM$_{2.5}$ have been
associated with health effects (e.g., Halonen et al 2009) and even mortality (WHO, 2013a; Raza et al.,
      2018). It is also biologically plausible that a threshold for the effects does not exist, due to the nature of
      the proposed physiological mechanisms of the effects, such as systemic inflammation (e.g., Lanki et al.,
      2015). However, in some recent global impact assessments, a lower cut-off concentration has been
      used (Gakidou et al., 2017).


      We have also made the simplification that the health effects of PM$_{2.5}$ were the same per mass unit for
      all emission source categories. The chemical composition of PM$_{2.5}$, and consequently the emission
      source, has been found to modify the health effects. For example, it has been suggested based on
      toxicological studies that secondary PM$_{2.5}$ may be less harmful that primary PM$_{2.5}$. However, the
current consensus is that the PM$_{2.5}$ sources cannot be ranked with respect to harmfulness, as the
      evidence is not sufficient for doing so (WHO 2013a, EPA 2009).

      Many of the health effects of PM$_{2.5}$ are lagged in time, whereas in the model all effects are treated as
      immediate ones. On one hand, the effect of the lag time is irrelevant, if the considered time scale is
very long. This is commonly the case for policy measures to curb PM$_{2.5}$ emissions; these are
      characteristically designed to be long-term solutions. On the other hand, the uncertainty of the cost
      estimates will increase over decades, as the population size and location, age structure, background
      risks and willingness to pay for better health will inevitably change.

The considered health outcomes have been presented in Table 1. These outcomes are mainly long-term
      effects. There is sufficient evidence also on the effects of short-term exposures on mortality, but as the
      short-term effects can be considered to be included in the estimates of the long-term effects, they were
      not explicitly included in the model. Regarding the restricted activity days, we did not include the days
      spent in a hospital (based on calculations on hospital admissions) or at home (calculations on lost work
days), to avoid double-counting.





Table 1. The considered health outcomes, age groups, types of exposure, risk ratios per concentration difference, their confidence intervals and annual background risks.

| Outcome | Age group | Exposure | Risk ratio per 10 µg/m³ | Confidence intervals, 95 % | Background risk, yearly |
|---|---|---|---|---|---|
| Mortality | > 30 yrs | $PM_{2.5}$ long-term | 1.062 | 1.040-1.083 | 1345,33 deaths/100,100 (2015) |
| Cardiovascular hospital admissions | All | $PM_{2.5}$ short-term | 1.0091 | 1.0017-1.0166 | 26,48/1000 (2014) |
| Respiratory hospital admissions | All | $PM_{2.5}$ short-term | 1.019 | 0.9982-1.0402 | 13.91/1000 (2014) |
| Neonatal infant mortality | 1-12 kk | $PM_{10}$ long-term | 1.04 | 1.02-1.07 | 0.77 deaths/1000 live births (2014); 10.12 births/1000 (2015) |
| Chronic bronchitis, incidence | >18 yrs | $PM_{10}$ long-term | 1.117 | 1.040-1.189 | 3.9 cases/1000 |
| Bronchitis, prevalence | 6-12 yrs | $PM_{10}$ long-term | 1.08 | 0.98-1.19 | 186/1000 |
| Work days lost | 20-65 yrs, at work | $PM_{2.5}$ 2 week | 1.046 | 1.039-1.053 | 9.85 days/ person (2008); employment rate 73,2 % (avg. 2011-2015) |
| Asthma symptoms, incidence | 5-19 yrs, asthmatics | $PM_{2.5}$ short-term | 1.028 | 1.006-1.051 | 35 asthmatics/1000; 17% of days with symptoms |
| Restricted activity days | ≥20 yrs | $PM_{2.5}$ 2 week | 1.047 | 1.042-1.053 | 19 days/ person |


The evidence for the concentration-response functions is stronger for mortality and hospital admissions, compared with the other health effects listed in Table 1. The concentration-response functions were nevertheless provided also for other health effects in the HRAPIE project. The causal
association for these effects can be considered to be probable; however, the magnitude of these effects cannot be precisely determined. We have included such effects in the model to avoid underestimation of the total health impacts. For the mortality, a risk ratio of 1.062 was used, which can be considered to be a state-of-the art value (e.g., Walton et al., 2015).



Some impacts of PM$_{2.5}$ have not been calculated in the total Finnish population, but in a specific age group. This selection was caused by the limitations of the epidemiological studies that provided the concentration-response functions. The HRAPIE project recommends computing the impact of PM$_{2.5}$ exposure on the restrictions of physical functioning without age limitations, although the original epidemiological study that provided the concentration-response function was conducted in a working-
age population (Ostro 1987). As a compromise, we have computed the impact in both working-age and elderly populations, but not in children, where the effect was considered to be too uncertain.

Concentration-response functions correspond to the relative effects of PM$_{2.5}$. In addition, information on the background risk is therefore needed for each outcome, to calculate the actual impact. In this
study, the background risk of mortality was obtained from Statistics Finland, and the information on hospital admissions from Eurostat. Other estimates of the background risk are based on previous EU-wide impact assessments (Hurley 2005, Holland 2014).

**2.4 Assessment of the economic impacts**

The economic cost values applied in the computations are presented in Table 2. The costs have been mainly selected according to the previous EU-wide impact assessments by Hurley et al. (2005) and Holland (2014). This also facilitates numerical comparisons with those studies. The mortality effects
have the largest impact on the total costs; the evaluation of the unit cost for mortality was therefore the most crucial parameter for the final results.

The monetized estimates in the computations of the economic impacts in this study are based on the average value of a life year (VOLY), instead of the value of statistical life (VSL). The VOLY-based
approach has been commonly used as a measure to assess a decrease in mortality risk (Im et al., 2018), whereas the VSL-based approach, despite its disadvantage, is in line with EPA's standard procedure and recommendations (Wolfe et al., 2019). VSL has been used in many studies in the U.S (i.e. Nahlik et al., 2016; Trejo-González et al., 2019; Wolfe et al. (2019), while VOLY has been mainly mentioned in EU researches (Im et al., 2018) It has been found that the VSL-based approach results in higher
economic cost values (e.g., EEA, 2014). The reason for this difference is that in the VSL approach, the increase in relative risk is uniformly applied to all the age groups, whereas in the VOLY approach, the relative risk is unequally distributed within the various age groups, resulting on the average in a smaller number of life years lost per case. However, some studies have adjusted for this factor (e.g., Muller and Mendehlson, 2009).

We have used both average and median values of VOLY in this study. However, the average value may correspond better to the willingness to reduce risks on a population level.






Table 2. The unit costs (in euros) of the health outcomes that were included in the model. All values are mainly based on the willingness to pay -approach.

| Outcome | Age group | Cost (€) | Additional information |
|---|---|---|---|
| Mortality, value of life | >30 years | 2.65 million | Average |
| Mortality, value of a life year | >30 years | 69000 (median) 160000 (average) | Median and average values |
| Cardiovascular hospital admissions | All | 2837 | The total sum consists of 628 €, the care costs for three days, 939 €, and the lost work days for five days, 1270 €. |
| Respiratory hospital admissions | All | 2837 | The total sum consists of 628 €, the care costs for three days, 939 €, and the lost work days for five days, 1270 €. |
| Chronic bronchitis, incidence | >18 years | 64500 | |
| Bronchitis, prevalence | 6-12 years | 784 | Cough symptom day 56 €, for 14 days |
| Lost work days | 20-65 years at work | 254/day | Working time 7.06 hours per day, the cost of each working hour 36 € |
| Restricted activity days | ≥20 years | 154/day | Based also on the cost of lost work days |


The unit cost of chronic bronchitis used in this study (200,000 €) was substantially lower than the corresponding value used in the previous EU-wide assessment by Hurley et al. (2005). The cost
estimate used here is based on the meta-analysis conducted in the HEIMTSA project; this new value has also been used in the most recent EU-wide assessment (Holland, 2014).

The cost of a hospital admission is partly based on the willingness to pay approach (WTP), as estimated by Ready et al. (2004). The WTP estimate takes into account three days in hospital care (because of a
respiratory disease), and five days of bed rest at home. In addition to WTP, direct health care costs (three days) and lost work days (five days) contribute to the total cost of a hospital admission. The health care cost estimate used in the calculations corresponds to the mean cost of an acute care admission (< 90 days) in primary care in Finland. Original unit cost has been adjusted for the year 2017 using data from Statistics Finland on the temporal changes of health care costs in Finland.




The estimated cost of a working day in Finland originates from 2012. The value has been adjusted for 2017 using the labor cost index reported by Statistics Finland. The cost of a restricted activity day consists of the cost of lost work days, and WTP costs of minor restrictions (symptoms) and more severe restrictions (bed rest at home). The WTP values are based on Ready et al. (2004). For the working age

population, it was assumed that 25 % of the restricted activity days were spent in bed at home, 25 % with symptoms at home, and 50 % at work with symptoms. Persons that are eligible for retirement (> 65 years, 25 % of the adult population) were assumed to spend 35 % of the restricted activity days in bed and the rest suffering from symptoms.

We adjusted the unit costs for inflation, but not for the changes in the income levels, which is accordance with the practice in the previous EU-wide assessment by Holland (2014). The WTP values were selected according to Ready et al. (2004), in which the results have been reported in pounds in 1998. In this study, these have been converted to euros using the purchasing power parity index, and to the values in 2017 using the Harmonised Index of Consumer Prices.


## 3  Results

### 3.1 Summary of the emissions of $PM_{2.5}$ and its main precursors in Finland


The total primary and main precursor emissions (for $NO_x$, $SO_2$ and $NH_3$) of $PM_{2.5}$ in Finland in 2015 have been presented in Fig. 1. Regarding the primary emissions of $PM_{2.5}$, the most important domestic pollution source categories were residential combustion (10.2 kt/a), and vehicular traffic and machinery (6.6 kt/a). The energy production and industrial combustion units, and industrial processes were

responsible for smaller proportions of the primary emissions of $PM_{2.5}$ (2.5 kt/a and 1.6 kt/a, respectively).





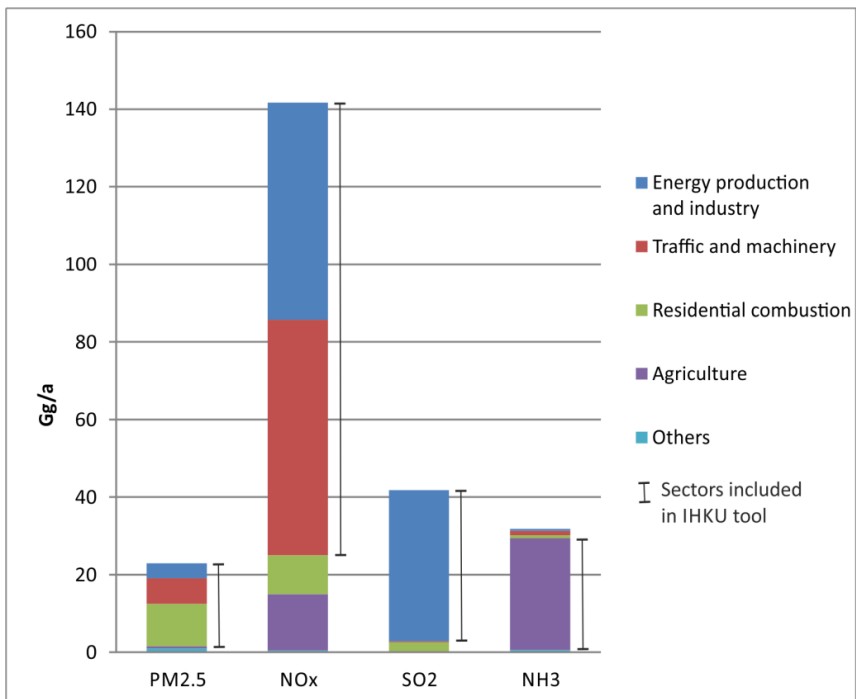


Fig. 1. The total amounts of the annual emissions in Finland in 2015 for the pollutants and source categories that were taken into account in this study (megatons/year). The $NO_x$ emissions were defined as the sum of NO and $NO_2$, presented as the mass of $NO_2$. The vertical bars show the emission source categories that were included in the simplified web-based tool of assessment.


Regarding the emissions of nitrogen oxides, vehicular traffic and machineries, and the energy production and industry were the most important source categories. The emissions of sulphur dioxide were mostly originated from energy production and industry, and the emissions of ammonia mostly

from the agricultural sector.

Karvosenoja (2008) has previously evaluated the uncertainties of the national annual average emission estimates of $PM_{2.5}$ for residential combustion and vehicular traffic. The estimates of uncertainties included both those for the use of fuels and for emission factors. The uncertainties were estimated to

range from − 36 % to + 50 % for residential combustion and from − 11 % to + 13 % for vehicular traffic, within 95 % confidence interval. The uncertainties of the emissions from point sources were found to be on the same level or lower as those for residential combustion. The uncertainties of the $PM_{2.5}$ precursor emissions were on the same level or lower than those for the primary $PM_{2.5}$ emissions.

Emissions from shipping have not been included in the above mentioned inventory. However, shipping emissions on a high resolution were used as input values in the SILAM model computations; described in more detail by Lehtomäki et al. (2018). The shipping emissions were provided by the computations using the STEAM emission model (e.g., Johansson et al., 2017).






### 3.2 The modelled changes of spatial concentration distributions caused by the changes of emissions

The atmospheric dispersion, and the changes of concentrations caused by the reductions of emissions, were evaluated for (i) vehicular traffic, (ii) working and off-road machinery, and (iii) small-scale residential combustion. The analyses were made separately for urban and rural areas. In addition, in case of residential wood combustion, we assessed separately the dispersion originated from (i) fireplaces and sauna stoves, and (ii) recreational houses and the boilers of detached houses.

The computations were made partly using the FRES model, partly using the SILAM model. The FRES model was mostly used for evaluating the reductions of concentrations caused by primary emissions (i.e., the emissions of $PM_{2.5}$). We used the FRES model for this purpose, as the spatial resolution was finer, compared with the SILAM model computations. The SILAM model was used for evaluating the reductions caused by the emissions of pollutants that form secondary particulate matter in the
atmosphere; the treatments of the FRES model do not include those processes.

The considered secondary pollutants in the following results include the most substantial ones for each source category; we did not evaluate the impacts of the complete range of secondary pollutants. The secondary pollutants included $NO_x$ originated from vehicular traffic and machineries, $NH_3$ from
agriculture, and $SO_2$ and $NO_x$ from power plants and industry. In addition, the SILAM model was used for evaluating the effects of the reductions of primary $PM_{2.5}$ originated from power plants and industry; this was done to achieve a better consistency of the predicted results with regard to the two considered secondary pollutants for this source category.


### 3.2.1 Vehicular traffic, working and off-road machinery, and residential wood combustion, evaluated using the FRES model

The predicted reductions of concentrations of $PM_{2.5}$ are presented in Figs. 2a-d for vehicular traffic,
and working and off-road machinery, separately for urban and rural areas. The computations were conducted using the FRES model on a spatial resolution of 250 x 250 $m^2$.



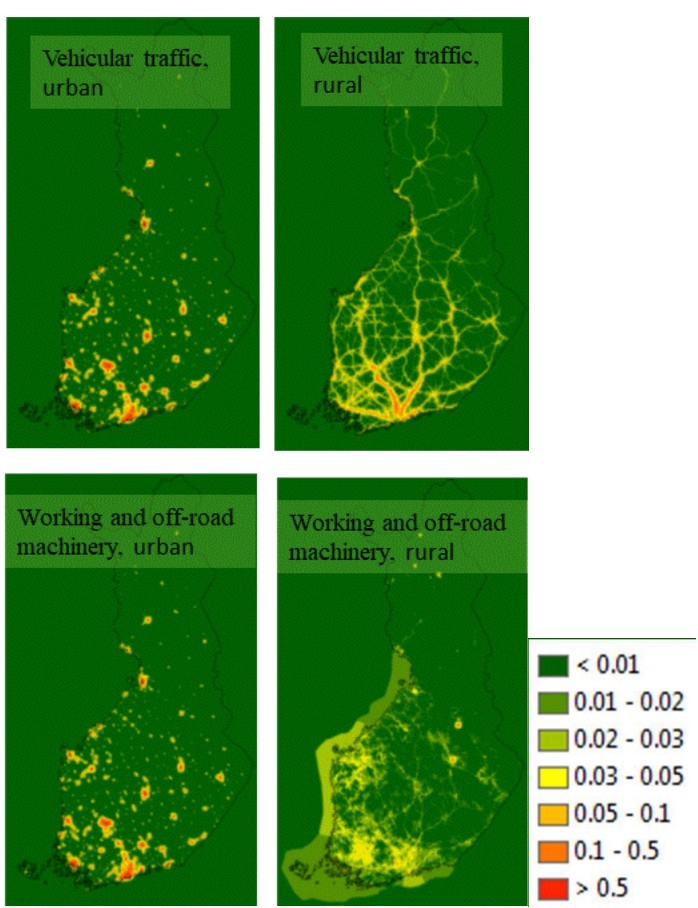


Figs 2a-d. The reductions of concentrations of $PM_{2.5}$ ($ng/m^3$), caused by a reduction of emissions of $PM_{2.5}$ by one ton. The results are presented for vehicular traffic (the upper panels), and working and off-road machinery (the lower panels). For both source categories, the changes in urban and rural areas are presented separately, in the left-hand side and right-hand side panels, respectively. The spatial
resolution is 250 x 250 $m^2$.

As expected, the urban reductions were focused on the largest urban agglomerations, cities and towns, for both source categories. The rural vehicular reductions were focused on the main road and street
network, especially in the most densely populated southern and western parts of the country. The machinery reductions were located within the most industrialized regions, most of which are located in south-western Finland; these were dispersed across a wider area, compared with the corresponding vehicular traffic reductions.

The predicted reductions of concentrations of $PM_{2.5}$ are presented in Figs. 3a-d for three segments of residential wood combustion. The reductions for fireplaces and sauna stoves are presented separately for urban and rural areas, whereas both the reductions in leisure homes and in the boilers of detached





houses are presented in one panel for the whole country. The computations were conducted using the FRES model on a spatial resolution of 250 x 250 m$^2$.


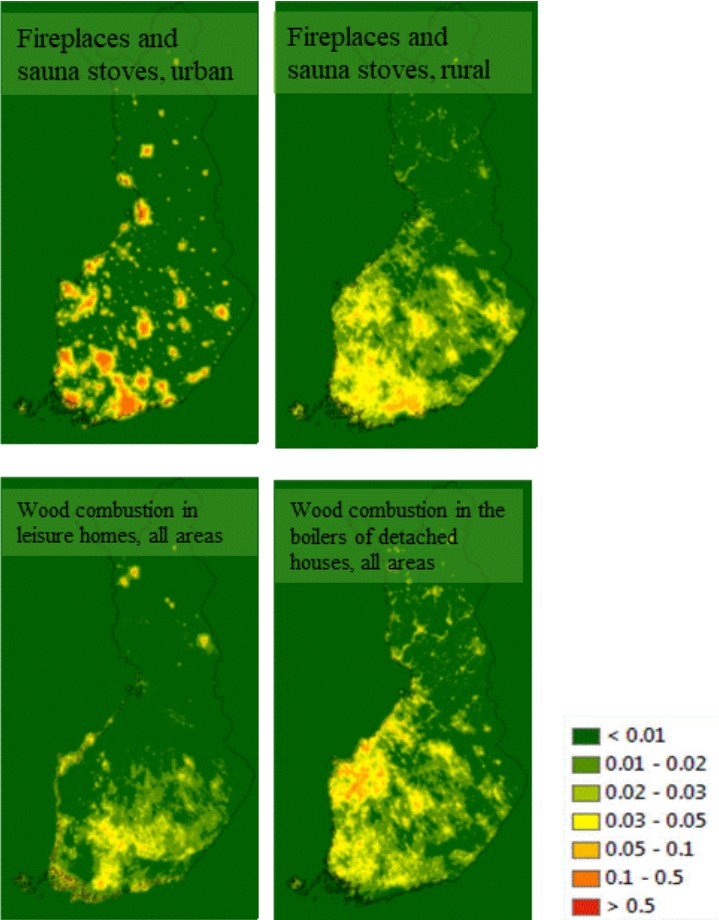

Figs 3a-d. The reductions of concentrations of PM$_{2.5}$ (ng/m$^3$), caused by a reduction of emissions of PM$_{2.5}$ by one ton, originated from small-scale residential combustion. The changes due to the emissions in fireplaces and sauna stoves are presented in the upper panels, and the changes due to the emissions in recreational houses and in the boilers of detached houses are presented in the lower panels. For both source categories, the changes in urban and rural areas are presented separately, in the left-hand side
and right-hand side panels, respectively. The spatial resolution is 250 x 250 m$^2$.

As expected, the urban reductions of the emissions for fireplaces and sauna stoves were focused on the largest urban agglomerations. However, the urban reductions in leisure homes and in the boilers of
detached houses were much more evenly distributed. The reductions for leisure homes are expectedly situated mostly in southern central Finland; this area has the most dense network of leisure homes. The



reductions from the boilers of detached houses are focused mostly in western Finland; this is caused by the differing cultural habits and preferences regarding housing in different parts of the country.

Kukkonen et al. (2018) has recently evaluated the uncertainties of the modelling system containing the urban scale UDM-FMI and CAR-FMI models. The model UDM-FMI was used for computing the source-receptor matrices within the FRES model. They evaluated the performance of the modelling system extensively, against the observations of $PM_{2.5}$ concentrations during 16 years at five measurement stations in the Helsinki Metropolitan Area. The uncertainties of the predicted annual
average concentrations of $PM_{2.5}$ ranged from – 18 % to + 15 %.

### 3.2.2 Vehicular traffic, machineries, agriculture, and power plants and industry, evaluated using the SILAM model


The reductions of $PM_{2.5}$ concentrations computed with the SILAM model are presented in Figs. 4a-e. The model grid covered entire Finland with a spatial resolution of 5 x 5 $km^2$.





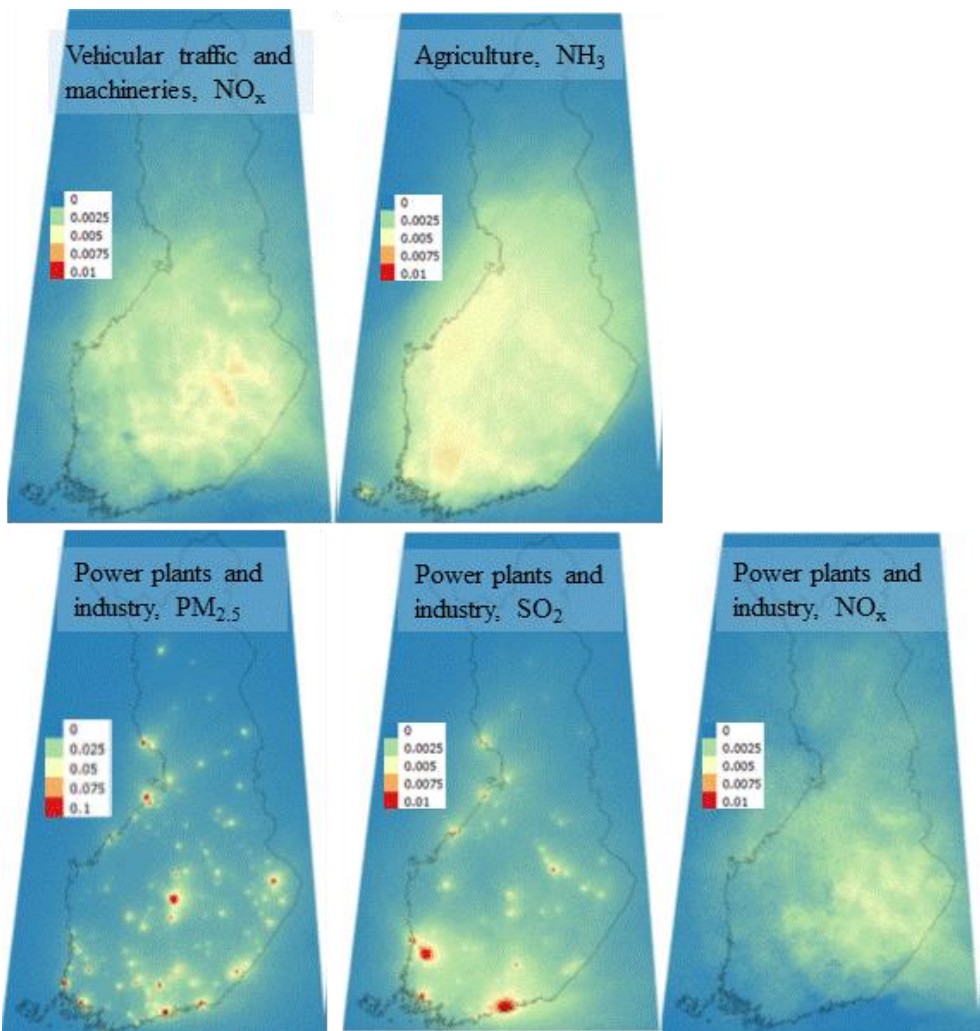


Figs 4a-e. The reductions of concentrations of $PM_{2.5}$ ($ng/m^3$), caused by a reduction of emissions of one ton of corresponding pollutants, originated from three source categories. The panels present the decrease in concentrations due to reduction of the following emissions: (a) $NO_x$ originated from vehicular traffic and machineries, (b) $NH_3$ originated from agriculture, (c) $PM_{2.5}$ originated from power plants and industry, (d) $SO_2$ originated from power plants and industry, and (e) $NO_x$ originated from power plants and industry. The spatial resolution is 5 x 5 $km^2$. The scale of reductions is different for the panel (c).


The most prominent $PM_{2.5}$ reductions in the close vicinity of the emitting sources were caused by the decrease of primary emissions of $PM_{2.5}$ originated from power plants and industry. A similar result was achieved for the $PM_{2.5}$ reductions caused by $SO_2$ emission reduction; however, the absolute values of





the reduction were an order of magnitude smaller, compared with those caused by the $PM_{2.5}$ emission reductions. This was caused partly by the fact that most of $SO_2$ is originated from relatively few major power plants and industrial regions, partly by the fairly slow chemical formation of sulphates.

The spatial patterns of the reduced $PM_{2.5}$ concentrations were more homogeneously distributed over
Finland in case of lowered emissions of the secondary pollutants NOx and $NH_3$. This was caused by the relatively longer time scales of the relevant chemical reactions and by the geographical locations of the main sources of NOx and $NH_3$, which are agricultural activities and vehicular traffic networks.

The predictions of the SILAM model have previously been extensively evaluated against monitoring
data. For most cases, there has been fairly good or good agreement, with a slight underestimation of PM concentrations (Prank et al., 2016). Most recently, Lehtomäki et al. (2018) have evaluated the accuracy of the SILAM model for predicting the annual average concentrations of $PM_{2.5}$. The predicted annual average values were on the average 5 % lower than the observations at 37 stations in Finland in 2015.

### 3.3 The health impacts

The health impacts were evaluated based on the atmospheric dispersion computations addressed in the
previous section. The impacts are presented in Table 3. The units of the values are different for the different columns. For instance, the values in the column 'mortality' are the numbers of the cases of premature deaths, and the values in the column 'Lost life years' are in years. The reported values are incremental health impacts, i.e., the presented impacts correspond to a unit amount (one kiloton) of emissions. The values are therefore not the total health impacts within the country.





Table 3. The health impacts caused by an emission of one kiloton of PM$_{2.5}$, NO$_x$, NH$_3$, or SO$_2$ in Finland in 2015, for various domestic pollution source categories in various regions. Notation: e.g., "NO$_x$ → secondary PM$_{2.5}$" refers to secondary fine particulate matter originated from the emissions of nitrogen oxides.


| Pollution source category | Mortality (cases) | Lost life years (years) | Chronic bronchitis (cases) | Bronchitis (cases) | Cardiovascular admissions (cases) | Respiratory admissions (cases) | Work days lost (days) | Restricted activity days (days) |
|---|---|---|---|---|---|---|---|---|
| Road transport, primary PM$_{2.5}$, urban | 108 | 660 | 118 | 305 | 49 | 54 | 41.7 10$^3$ | 15.8 10$^3$ |
| Road transport, primary PM$_{2.5}$, non-urban | 11 | 64 | 11 | 36 | 5 | 5 | 3.86 10$^3$ | 1.46 10$^3$ |
| Non-road & machinery, primary PM$_{2.5}$, urban | 132 | 823 | 146 | 357 | 60 | 66 | 52.1 10$^3$ | 19.6 10$^3$ |
| Non-road & machinery, primary PM$_{2.5}$, non-urban | 4 | 24 | 4 | 13 | 2 | 2 | 1.36 10$^3$ | 0.53 10$^3$ |
| Residential houses, wood stoves and saunas, primary PM$_{2.5}$, urban | 54 | 331 | 57 | 178 | 24 | 27 | 20.0 10$^3$ | 7.63 10$^3$ |
| Residential houses, wood stoves and saunas, primary PM$_{2.5}$, non-urban | 7 | 42 | 7 | 22 | 3 | 3 | 2.30 10$^3$ | 0.90 10$^3$ |
| Recreational houses, wood stoves and saunas, primary PM$_{2.5}$ | 4 | 26 | 4 | 14 | 2 | 2 | 1.55 10$^3$ | 0.60 10$^3$ |
| Residential houses, wood boilers, primary PM$_{2.5}$ | 9 | 56 | 9 | 31 | 4 | 4 | 3.16 10$^3$ | 1.27 10$^3$ |
| Road transport, NOx → secondary PM$_{2.5}$ | 1 | 4 | 1 | 2 | 0.3 | 0.3 | 0. 22 10$^3$ | 0.086 10$^3$ |
| Agriculture, NH$_3$ → secondary PM$_{2.5}$ | 0.9 | 6 | 1 | 3.1 | 0.41 | 0.46 | 0.33 10$^3$ | 0.13 10$^3$ |
| Industry and power plants, primary PM$_{2.5}$ | 7.2 | 44 | 7.5 | 23 | 3.19 | 3.50 | 2.63 10$^3$ | 1.01 10$^3$ |
| Industry and power plants, SO$_2$ → secondary PM$_{2.5}$ | 1 | 6 | 1.1 | 3.4 | 0.46 | 0.51 | 0.38 10$^3$ | 0.15 10$^3$ |
| Industry and power plants, NO$_x$ → secondary PM$_{2.5}$ | 0.4 | 2 | 0.4 | 1.1 | 0.15 | 0.17 | 0.12 10$^3$ | 0.049 10$^3$ |


In general, the impacts were largest in case of primary PM$_{2.5}$ emissions, compared with those for the corresponding secondary pollution. As expected, the impacts in urban areas were also substantially larger than the corresponding impacts in non-urban areas. Regarding the pollution source categories, the most important were non-road and machinery, road transport in urban areas, and wood stoves and

saunas in residential houses.

In addition to the above-mentioned health impacts, the infant mortality and the asthma symptoms were also considered. However, these impacts were negligible compared with other considered impacts. In case of the infant mortality, the background risk was very low, and for the asthma symptoms, both the

prevalence and risk ratio were low. The infant mortality and asthma were therefore excluded from further analysis.



The uncertainty of the health effect values can be estimated based on the adopted concentration-response functions. The majority of the public health costs are related to premature mortality. We therefore address here the average concentration-response for the $PM_{2.5}$ related to mortality, which has been assumed to be 1.062 (cf. Table 1), with a linear dependency with respect to the concentration. The 95 % confidence limits of this value range from 1.040 to 1.083. We therefore conclude that the lowest and highest health effect estimates (within the 95% confidence interval) could be approximated by multiplying by the predicted health effect values by 0.65 (i.e., 4.0 % / 6.2 %) and 1.3 (8.0 % / 6.2 %).

### 3.4 The economic impacts

We have assessed the economic impacts of the selected potential PM2.5 emission reductions, based on the health impacts addressed in the previous section. These have been computed for a change of one ton of the annual emissions for the selected pollutants in 2015. The results include only the impacts of the Finnish emissions to the population in Finland; i.e., the health impacts caused by the Finnish emissions in other countries have not been evaluated.

First, the estimated contributions to the total costs were evaluated for the various health outcomes. The detailed results of these computations are presented in Appendix A. The mortality effects were clearly the largest factor affecting the total costs. However, also the costs associated with restricted activity days, lost working days and chronic bronchitis were found to be substantial.

The final results of the economic cost computations are presented in Table 4. The values in Table 4 have been presented for three alternative options for computing the economic impacts: (i) the average value of life year (VLY), assumed to be equal to 160 000 €, (ii) the median value of life year, assumed to be 69 000 €, and (iii) the average value of statistical life (VSL), assumed to be 2.65 million €.

The results have been presented separately for the source categories that have relatively lower and higher emission heights, respectively. The latter category includes the industrial pollution sources and power plants. The results have also been presented in terms of the pollutant, the emissions of which have been assumed to be decreased; these include both primary $PM_{2.5}$ and the main precursor substances. For the most significant source categories and substances, the results have also been presented separately for various types of areas, such as the urban and rural areas. For the primary $PM_{2.5}$ emissions from industry and power plants, the results were presented separately for areas with different population densities; these included the capital area, and relatively more and less densely populated municipalities, respectively.

There are substantial variations of the results, depending on the economic computation methods (average or median VLY or VSL). However, the order of these results is the same for all the results; e.g., the computation using VSL results in the highest economic values. We have therefore illustrated the results computed with one of these methods, i.e., the average of VLY, in Figs. 5a-b and 6a-b. These figures therefore can be used to illustrate the relative economic benefits of the selected emission reduction alternatives.

The economic benefits are clearly largest for the emission reductions for the source categories that have low emission heights (Figs. 5a-b), compared with those with substantial emission heights (Figs 6a-b). For both kinds of source categories, the reductions are expectedly substantially more effective in the





more densely populated regions. For instance, the reductions of the $PM_{2.5}$ emissions originated from vehicular traffic, non-road and machinery and residential wood combustion in urban areas result in approximately an order of magnitude higher economic benefits, compared with the impacts of the corresponding emission reductions in rural areas. The results also show that the reduction of the precursor emissions of $PM_{2.5}$, such as $NO_x$, $NH_3$ and $SO_2$ was clearly less effective for reducing both the $PM_{2.5}$ concentrations and the adverse economic impacts, compared with reducing directly the
emissions of $PM_{2.5}$.

The uncertainties of the economic evaluations can be estimated based on the difference of the three alternative methods, i.e., those based on the average and median VLY, and the one based on the average VSL. Assuming that the average VLY would be the base value (denoted here as 1.0), the
uncertainty of this estimate would range from 0.57 to 2.2.






Table 4. Economic benefits obtained by the assumed reductions of emissions, in thousand euros per ton of emissions. The results are presented for the various source categories, in various domains. The first presented value has been computed based on the average value of life year, and the two values in parenthesis based on the median value of life year and the average value of statistical life, respectively.


| Source category and the emission height | Region, in which the reduction of emission takes place | | |
|---|---|---|---|
| **Emissions at low height** | **Urban area** | **Non-urban area** | |
| Road transport, primary PM$_{2.5}$ | 140 (80–320) | 13 (7.6–31) | |
| Non-road and machinery, primary PM$_{2.5}$ | 170 (100–390) | 5.0 (2.8–11) | |
| Residential houses, wood and sauna stoves, primary PM$_{2.5}$ | 70 (40–160) | 8.7 (4.8–19) | |
| | **Whole of Finland** | | |
| Recreational houses, wood stoves and sauna stoves, primary PM$_{2.5}$ | 5.5 (3.1–13) | | |
| Residential houses, wood boilers, primary PM$_{2.5}$ | 12 (6.6–27) | | |
| Road transport, NO$_x$ emissions forming secondary PM$_{2.5}$ | 0.82 (0.46–1.8) | | |
| Agriculture, NH$_3$ emissions forming secondary PM$_{2.5}$ | 1.2 (0.70–2.8) | | |
| **Emissions at substantial height** | **Helsinki capital area** | **Municipalities with > 50 000 inhabitants** | **Other areas** |
| Industry and power plants, primary PM$_{2.5}$ | 20 (11–44) | 6.9 (3.9–16) | 5.4 (3.1–12) |
| | **Whole of Finland** | | |
| Industry and power plants SO$_2$ emissions forming secondary PM$_{2.5}$ | 1.3 (0.73–3.1) | | |
| Industry and power plants, NO$_x$ emissions forming secondary PM$_{2.5}$ | 0.43 (0.24–1.0) | | |





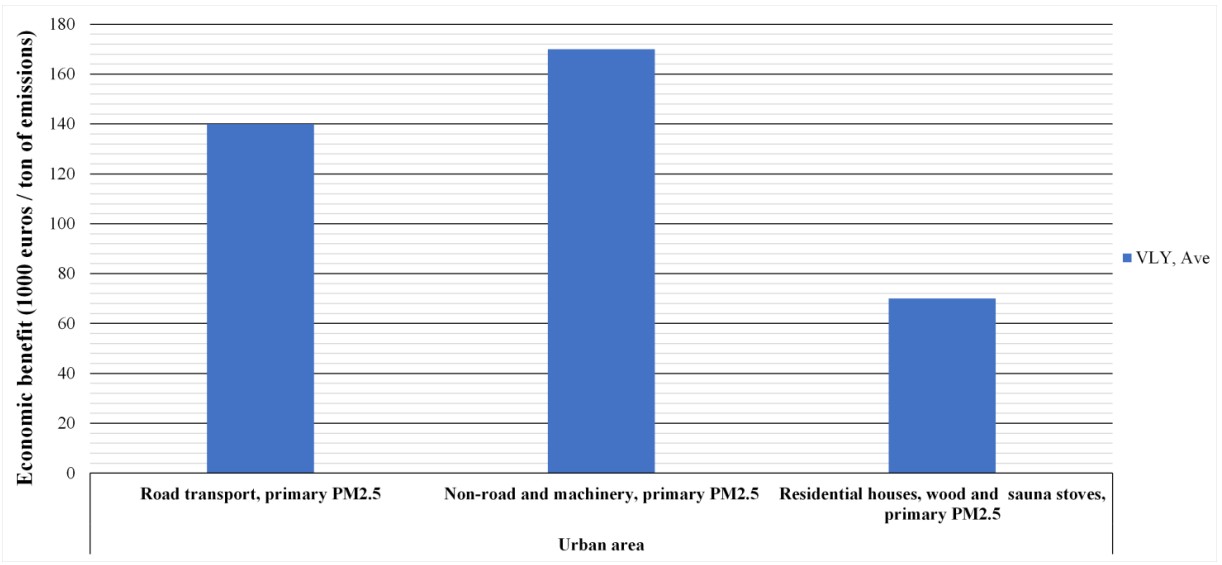

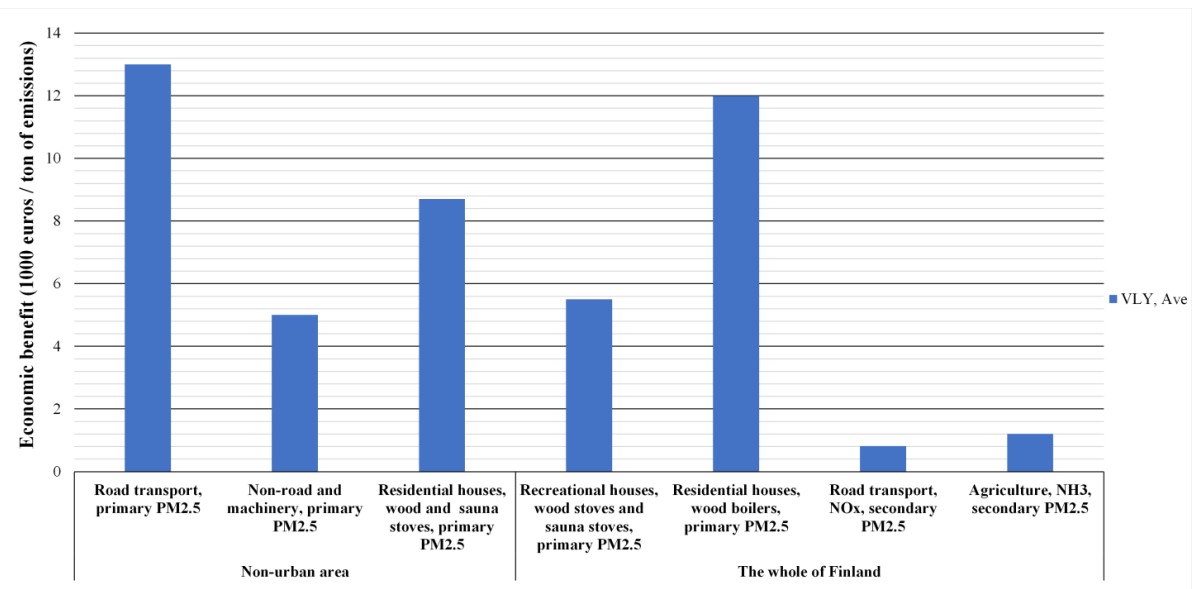


Figs 5a-b. Economic benefits obtained by the assumed reductions of emissions, in thousand euros per ton of emissions, for sources having a low emission height. The results are presented for urban areas (upper panel, a), and for non-urban areas and for the whole country (lower panel, b) in case of various source categories and pollutants, in various domains. All the values correspond to the computations using the average value of life year.




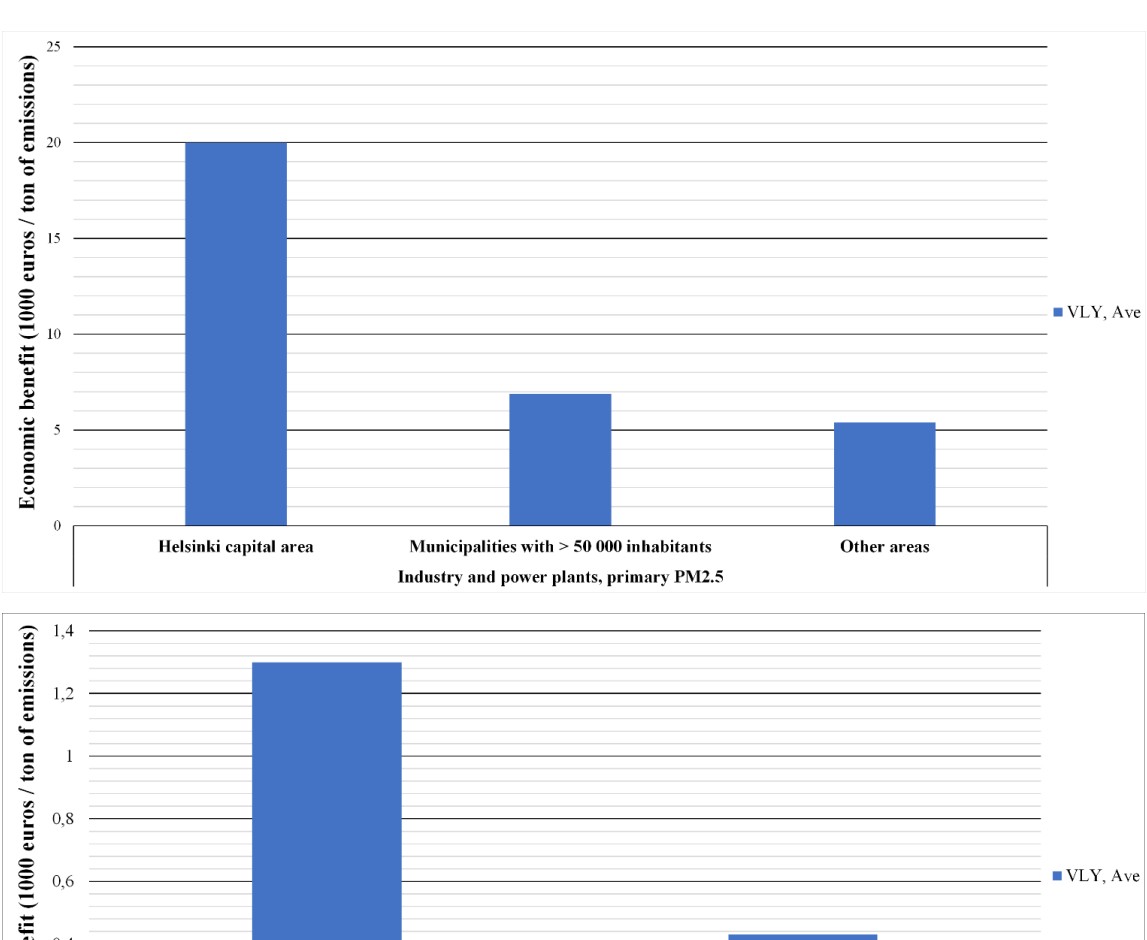


Figs 6a-b. Economic benefits obtained by the assumed reductions of emissions, in thousand euros per ton of emissions, for sources having a high emission height. These results are presented for industry and power plants. The upper panel (a) presents results for the emission reductions of $PM_{2.5}$ for various geographic regions, and the lower panel (b) for the emission reductions of $SO_2$ and $NO_x$ for the whole of Finland. All the values correspond to the computations using the average value of life year.






### 3.4 An open access tool of assessment for evaluating the economic impacts


We have also designed and implemented a user-friendly internet-based tool of assessment for evaluating the health costs of various assumed emission reduction options. This tool was designed to facilitate an easy use of the model to policy makers, stakeholders and environmental experts. The tool can be accessed via a user-friendly interface in the internet
(https://wwwp.ymparisto.fi/IHKU/haittakustannuslaskuri/). This calculator is based on the numerical results of this study (such as those presented in Table 4); however, some minor simplifications were made regarding the included emissions. The included emission source categories have been presented by the vertical bars in Fig. 1.

The internet-based tool requires as input value the amount of reduced emissions (tons/year) for a source category, pollutant and region, corresponding to a selected abatement measure, bundle of measures or strategy. The tool can then be used to compute as output the annual financial benefits of the measure or strategy (in euros), presented both tabulated and graphically. For instance, if the policy maker has an estimate of (i) the emission reduction that could be achieved by a potential abatement measure and (ii)
the economic cost of implementing the measure, he or she can use the tool to analyze whether the measure would result in more substantial economic benefits, compared with the costs. Clearly, the tool can also be used for comparing the cost-effectiveness of alternative potential emission reductions.

## 4 Conclusions

We have presented an integrated tool of assessment for evaluating the public health costs of fine particulate matter ($PM_{2.5}$) in ambient air. The model was applied to analyze the costs of the domestic primary and precursor emissions of $PM_{2.5}$ in Finland in 2015. The model does not address other effects
of fine particulate matter in ambient air, such as, e.g., the impacts on climate change and on the state of the environment.

We have evaluated the national emissions on a high spatial resolution, 250 x 250 $m^2$ for the whole country. The atmospheric dispersion was analyzed both using a chemical transport model (SILAM) and
a decision-support tool that uses source-receptor matrices (FRES). The health and economic impacts were analyzed based on the most significant health outcomes. The risk ratios and economic evaluations were based on the most recent results in the literature. However, reliable concentration-response functions were available only for a limited number of health outcomes. For example, the effects of the long-term exposure on the cardiorespiratory and cancer morbidity could not yet be included in the
model, although these can be associated with substantial health care and willingness to pay costs. The economic costs of the $PM_{2.5}$ exposures have therefore probably been under-predicted in this respect.

There are also substantial uncertainties in quantifying the economic effects of the various health outcomes. In particular, the final estimates of the economic costs substantially depend on the selection
of the economic measures; these can alternatively be the value of life year, either as an average or a median, or the value of statistical life. We have therefore presented three potential values for each public health cost, for each source category and pollutant.




The total uncertainties of the adopted impact pathway approach can be analyzed by studying the uncertainties for each of the stages of the assessment. The largest uncertainties to the final cost estimates were caused by the health impact assessments and the economic evaluations. We evaluated that the lowest and highest health effect estimates (within the 95% confidence interval) ranged from 0.65 to 1.3 (when the predicted optimal evaluation is normalized to 1.0). Similarly, the uncertainty of the economic cost estimate was found to range from 0.57 to 2.2. The uncertainty of the assessment resulting from these two main sources of uncertainty would therefore vary approximately from 0.36 to 2.9.

The developed modelling system can be used to evaluate the costs of the health damages for various emission source categories, for a metric ton of emissions of $PM_{2.5}$. The economic benefits were clearly largest for the emission reductions for the source categories that have low emission heights, such as vehicular traffic, non-road and machinery, and residential wood combustion. For all source categories, the emission reductions were substantially more effective, even by an order of magnitude, in the urban areas, compared with those in rural areas. The reduction of the precursor emissions of $PM_{2.5}$ was clearly less effective, compared with reducing directly the emissions of $PM_{2.5}$.

Based on the results achieved in this study, we have designed an open-access, user-friendly web-based tool of assessment. Both the final results obtained in this study, and the web-based assessment tool can be used in analyzing the economic benefits associated with various alternative abatement measures, policies or strategies. The models and the numerical results can also be used to inter-compare the cost-efficiency of different potential emission mitigation measures and strategies.

## 5. Code and data availability

The SILAM code is publicly available.

The emission data, and the predicted concentration data used in this study is available, by contacting the responsible authors, i.e., N. Karvosenoja, J. Kukkonen and M. Sofiev.

## 6. Acknowledgements

We acknowledge the funding of the Government of Finland for the project 'Air Pollution Damage Cost Model for Finland (IHKU)', within the research programme 'Bioeconomy and clean solutions'. We especially wish to thank the supervisor of the project Ms. Sirpa Salo-Asikainen (Ministry of the Environment). This work was also partly funded by the Academy of Finland, the project 'Global health risks related to atmospheric composition and weather (GLORIA)', by Nordforsk, the project 'Understanding the link between Air pollution and Distribution of related Health Impacts and Welfare in the Nordic countries (NordicWelfAir)', and by European Union within the Horizon 2020 programme, project 'Exposure to heat and air pollution in EUrope – cardio-pulmonary impacts and benefits of mitigation and adaptation (EXHAUSTION)'.





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



## Appendix A. Public health costs for various health outcomes

Table A1. Public health costs in euros of one ton of source-specific $PM_{2.5}$ emissions for various health outcomes in Finland in 2015. Notation: VOLY = value of life year, VSL = value of statistical life. The values have been presented using three significant numbers.

| Emission source category, pollutant and the region | Mortality, $10^3$ VOLY average | Mortality, $10^3$ VOLY median | Mortality, $10^3$ VSL average | Chronic bronchitis | Bronchitis | Cardiovascular admissions | Respiratory admissions | Work days lost | Restricted activity days |
|---|---|---|---|---|---|---|---|---|---|
| Road transport, primary $PM_{2.5}$, urban | 106 | 45.5 | 286 | 7590 | 239 | 139 | 152 | 10600 | 15800 |
| Road transport, primary $PM_{2.5}$, non-urban | 102 | 4.42 | 27.8 | 702 | 28 | 13 | 15 | 982 | 15800 |
| Non-road & machinery, primary $PM_{2.5}$, urban | 132 | 56.8 | 349 | 9420 | 280 | 171 | 187 | 13200 | 19600 |
| Non-road & machinery, primary $PM_{2.5}$, non-urban | 3.84 | 1.66 | 10.2 | 252 | 10 | 5 | 5 | 346 | 527 |
| Residential houses, wood stoves & sauna, primary $PM_{2.5}$, urban | 53.0 | 22.8 | 143 | 3660 | 139 | 69 | 76 | 5080 | 7630 |
| Residential houses, wood stoves & sauna, primary $PM_{2.5}$, non-urban | 6.72 | 2.90 | 17.4 | 429 | 18 | 8 | 9 | 583 | 900 |
| Recreational houses, wood stoves & sauna, primary $PM_{2.5}$ | 4.16 | 1.79 | 11.4 | 287 | 11 | 5 | 6 | 394 | 601 |
| Residential houses, wood boilers, primary $PM_{2.5}$ | 8.96 | 3.86 | 24.4 | 601 | 24 | 11 | 13 | 803 | 1270 |
| Road transport, NOx → secondary $PM_{2.5}$ | 0.64 | 0.28 | 16.4 | 41 | 2 | 1 | 1 | 55 | 86 |
| Agriculture, NH3 → secondary $PM_{2.5}$ | 0.96 | 0.41 | 2.50 | 62 | 2 | 1 | 1 | 85 | 131 |
| Industry&power plants, primary $PM_{2.5}$ | 7.04 | 3.04 | 19.1 | 483 | 18 | 9 | 10 | 667 | 1010 |
| Industry&power plants SO₂ → secondary $PM_{2.5}$ | 0.96 | 0.41 | 2.77 | 70 | 3 | 1 | 1 | 97 | 146 |
| Industry&power plants, NOx → secondary $PM_{2.5}$ | 0.32 | 0.14 | 0.93 | 23 | 1 | 0 | 0 | 31 | 49 |





## 8. Author contributions

Jaakko Kukkonen has compiled and written a substantial part of the article. N. Karvosenoja, T. Lanki and J. Kukkonen have proposed this study for funding and written a research plan. M. Savolahti, V.-V. Paunu and N. Karvosenoja have done the emission and dispersion computations with the FRES model, part of the economic computations, compiled a substantial fraction of the results together, and written part of the article. Y. Palamarchuk and M. Sofiev have conducted the SILAM computations and written the corresponding parts of the article. T. Lanki and P. Tiittanen have conducted the health impact assessments. V. Nurmi has done part of the economic computations, contributed to the section on economic assessments, and written a substantial part of the literature review in the introduction. L. Kangas and A. Karppinen have compiled the required meteorological information and evaluated the dispersion matrices for the FRES model. Ms. A. Maragkidou has post-processed the data and contributed to the writing of the literature review and other parts of the article.

## 9. Competing interests

The authors declare that they have no conflict of interest.