# Peer review of "Modelling of the public health costs of fine particulate matter and results for Finland in 2015"

_Atmospheric Chemistry and Physics, 2019_

## Referee Comment (RC1) · Anonymous Referee #1 · 12 Nov 2019

The paper aims at estimating monetized health impacts caused by emissions of 1 t of PM2.5 or 1 t of PM2.5 precursors. The monetized health impacts are given distinguishing between emissions of main source categories and emissions in urban vs. non-urban areas. The calculation is made according to the state of science; the science used is sound, the methodology is well described, calculations have been thoroughly made. However, progress beyond the current state is not made, new developments are not addressed. An example for currently analysed improvements of the methodology is accounting for NO2 impacts, which the WHO now considers as likely, though less certain than PM2.5 impacts. Another issue is using exposure (i.e. concentration where people are) instead of concentration in the background, which would involve analysing also indoor sources (smoking, frying, wood heating). Another field
is source apportionment to improve the large uncertainty of atmospheric modelling. In the text, a small chapter should at least mention current work on improving the methodology. A new part is the calculation of health impacts for emissions of different source groups like transport, residential a.s.o. Up to now, monetized health impacts have been estimated differentiated according to height of release and urban/non-urban area (see e.g. http://ecoweb.ier.uni-stuttgart.de/EcoSenseLE/current/index.php ). Some important (now older, but still valid) guidelines for the methodology could be cited, e.g. the ExternE Externalities of Energy Methodology 2005 Update; downloadable at http://www.externe.info/externe_d7/?q=node/30 or the IEHIAS Integrated Environmental Health Impact Assessment System (2011); web based guidance system; accessible under www.integrated-assessment.eu . The result is indeed useful for consultants or decision makers in Finland, that want to identify the most efficient measures for reducing health effects from fine particles. They may use the given unit cost figures for estimating the health benefit caused by reduced emissions, and could also use the tool provided. The editors should decide, whether the publication fits into the scope of their journal or whether a journal more oriented towards environmental policy application might be a better choice.
* * *

---

## Referee Comment (RC2) · Ståle Navrud (Referee) · 12 May 2020

This paper is an excellent and state-of-the-art application of the impact pathway approach (also termed the damage cost approach by economists) to assess public health costs of fine particulate matter (PM2.5) in ambient air from domestic primary and precursor emissions of PM2.5 in Finland in 2015. The paper provides a very transparent overview of the assumptions and calculations in all steps of the impact pathway model they develop; from emissions from different sources through dispersion and exposure to concentration-response functions and the economic valuation of the selected health endpoints. In the economic valuation of premature death due to PM2.5 the authors argue for using the number of lost life years and the Value of a Life Year (VOLY) rather than valuing all people at the Value of Statistical Life (VSL). This implies assigning a

lower value to the elderly, that make up the majority of people dying prematurely due to PM2.5. The authors state in line 402-407: "The monetized estimates in the computations of the economic impacts in this study are based on the average value of a life year (VOLY), instead of the value of statistical life (VSL). The VOLY-based approach has been commonly used as a measure to assess a decrease in mortality risk (Im et al., 2018), whereas the VSL-based approach, despite its disadvantage, is in line with EPA's standard procedure and recommendations (Wolfe et al., 2019)". The "despite disadvantages "phrase should be deleted. They should instead say choosing between the VSL and VOLY approaches is also an ethical question in terms of whether one should assign the same economic value to all adults, independent of age; and that a constant VOLY assumes that people value a life year the same independent of age. Also the authors further down in the paper (line 669-672) say that they have performed calculations using both VOLY and VSL (see also table 4), in order to show the difference in calculated economic benefits from emission reductions; which is of course very good that they do. However, in line 402 (see citation above) they state that they only used VOLY. This has to be corrected to made consistent with the calculations/sensitivity analyses they actually performed. In line 399 after the sentence "This also facilitates numerical comparisons with those studies" they should add that: "Ideally a country-specific VSL (and a derived VOLY from this VSL) for Finland should be applied in this analysis, which could be based on value transfer (Navrud and Ready 2008) from the most recent global meta analysis of stated preference studies of VSL (Lindhjem et al 2011). However, this would preclude the direct comparison of results with similar impact pathway models e.g. Holland et al (2005). In the appendix the authors use VOLY as the abbreviation for Value of a Life Year , whereas in the text also VLY is used (e.g. in the paragraphs including lines 670, 685 and 705). This should be consistent, and I suggest using VOLY all the way through the paper. They find that the economic benefits om emission reductions are clearly largest for the emission reductions for the source categories that have low emission heights, and are located in more densely populated regions (i.e. vehicular traffic, non-road and machinery and

residential wood combustion urban versus rural areas). In the Abstract they say "It was found that economically the most effective measures would be the reduction of the emissions in urban areas of (i) road transport, (ii) non-road vehicles and machinery, and (iii) residential wood combustion". However, in economics cost-effectiveness, that this can statement can interpreted as, is a measure of e.g. reduced number of kg of PM 2.5 emissions per euro of abatement costs. Thus, the authors should rather say that " the economic benefits in terms of avoided public health costs is largest for measures that will reduce of (i) road transport, (ii) non-road vehicles and machinery, and (iii) residential wood combustion". Whether these measures have the largest net benefits (Economic health benefits minus the abatement costs) depends of course on the costs of reducing emissions from theses sources. This also has to be corrected when using the word "effective" both in the abstract and in line 399-805.

There could also be other economic benefits than to public health from reducing the PM2.5 emissions , such as reduced soiling and corrosion of residential and commercial buildings, historic monuments and cultural heritage buildings as well as reduced environmental damages., The authors very briefly mentions this at the end of the paper, but they could also mention that there are ways to value these economic benefits and refer to e.g. Navrud and Ready (2002, 2007) and Watt et al (2009).

References: Holland, M, A. Hunt, F. Hurley, S. Navrud, and P.Watkiss 2005: Methodology for Cost-Benefit Analysis of CAFE (Clean Air for Europe). Volume 1: Overview and methodology. Report to DG Environment. European Commission, 112pp. Lindhjem, H., S. Navrud, N.A. Braathen, and V. Biausque 2011: Valuing lives saved from environment, transport and health policies. A meta analysis. Risk Analysis 31 (9); 1381-1407. Navrud, S. and R. Ready (eds.) 2002: Valuing Cultural Heritage. Applying environmental valuation techniques to historical buildings, monuments and artifacts. Edward Elgar Publishing. Navrud, S and R. Ready (eds.) 2007: Environmental Value Transfer: Issues and Methods. Springer, Dordrect, The Netherlands.

Watt, J., S. Navrud, Z. Slizkova and T. Yates 2009: Economic Evaluation.. Chapter

7 (pp. 189-214) In Watt, J., J. Tidblad, V. Kucera and R. Hamilton (eds.) 2009: The Effects of Air Pollution on Cultural Heritage. Springer. 306 pp.
* * *

---

## Author Response (AR2)

The comments of both referees are on black font below, and the authors' response in on blue.

**Referee # 1 (anonymous)**

**Referee's comment**

The paper aims at estimating monetized health impacts caused by emissions of 1 t of PM2.5 or 1 t of PM2.5 precursors. The monetized health impacts are given distinguishing between emissions of main source categories and emissions in urban vs. non-urban areas. The calculation is made according to the state of science; the science used is sound, the methodology is well described, calculations have been thoroughly made.

**Authors' response**

Thank you for these positive comments.

**Author's changes in manuscript**

No changes made.

**Referee's comment**

However, progress beyond the current state is not made, new developments are not addressed.

**Authors' response**

The manuscript includes also progress beyond the current state-of-the-art. However, we have not highlighted these sufficiently well in the original manuscript.

First, the emission and partly also dispersion modelling has been performed using a very fine spatial resolution, up to 250 x 250 m$^2$; these have not previously been used for such a geographically extensive area. In comparison, e.g., Heo et al. (2016) used as the basic material the predicted concentrations on a spatial resolution of 36 x 36 km$^2$, i.e., two orders of magnitude coarser than in the present study. This resolution scale is characteristic of all the previous studies on public health costs.

We have previously shown (Karvosenoja et al., 2010, Korhonen et al., 2019) that the exposure values evaluated using integrated assessment models can be very sensitive to the methodology. In particular, exposure values can substantially increase with an increasing spatial resolution. Karvosenoja et (2010) showed that using a finer spatial resolution, from 10 km to 1 km, resulted in an increase of the population weighed concentration caused by traffic emissions by an order of magnitude. It is therefore essential to use a sufficiently fine model resolution in view of the health impacts. This is especially important for primary particles from emission sources at fairly low emission heights.

Second, we have also considered the health costs related to the PM$_{2.5}$ precursor emissions on a much finer spatial resolution (5 x 5 km$^2$) than any previous corresponding study. A couple of previous studies have considered precursors (Muller and Mendehlson, 2009, Heo et al. 2016). However, both of the above mentioned studies addressed the U.S. as their modelling domain, Heo et al. (2016) on a resolution of tens of km. The resolution used in the study of Muller and

Mendehlson (2009) was not reported, but considering their results suggests a resolution of tens of km. Air chemistry and the formation of particulate matter is strongly dependent on the chemical composition of the emitted pollutants and climatic factors; the above mentioned American studies can therefore not per se be generalized for European conditions. In addition, we have also allowed for the organic PM in our computations; this was not taken into account by Muller and Mendehlson (2009) or Heo et al. (2016).

Third, we have also programmed an internet-based computation tool, which is publicly available. We expect that this concept and the methods included within this easy-to-use tool would be useful also internationally.

Fourth, the manuscript includes a thorough, up-to-date literature review in this field (in the introductory section). Such a detailed review has not previously been presented in the literature. We have also reviewed in detail the current methods for evaluating the health impacts (in section 2.3) and economic impacts (section 2.4).

Although several public health cost studies with similar aims have been conducted in the U.S., reviewed European articles on health costs are much more scarce. Such studies will need to be done and published for several European regions, for various pollutants, and using different methodologies.

This is needed, as the relevant environmental conditions, such as air chemistry, population densities, the structure of emission source categories, the economic costs of life years, and many other relevant factors, are substantially different in various parts of Europe. The range of uncertainty in such estimates is also large, as shown also in the present study. For consolidating such estimates for various regions, pollutants, methods, etc., therefore requires that such evaluations will be presented for several European regions.

**Author's changes in manuscript**

We have restructured and revised the conclusions section; the revised conclusions contains a more explicit and clear description on what new developments were introduced in this study, in the context of previous literature.

**Referee's comment**

An example for currently analysed improvements of the methodology is accounting for NO2 impacts, which the WHO now considers as likely, though less certain than PM2.5 impacts.

**Authors' response**

This is indeed a relevant and timely comment. Especially older diesel cars emit significant amounts of oxides of nitrogen. As diesel cars have become more common, there has been an intense debate about the need to curb $NO_2$ emissions in Europe. Long-term exposure to $NO_2$ has been associated with increased mortality in many epidemiological studies.

However, separation of the effects of $NO_2$ from the effects of particulate matter is challenging, due to their high correlation. Because of major uncertainties concerning concentration response functions, we did not include $NO_2$ in this study. WHO has also acknowledged that the effects of $NO_2$ are partly overlapping with $PM_{2.5}$ effects in epidemiological studies (WHO, 2013).

**Author's changes in manuscript**

We have added a clarification on why the direct health impacts of NO2 were not addressed in this study, to the second last paragraph of the introductory section.

**Referee's comment**

Another issue is using exposure (i.e. concentration where people are) instead of concentration in the background, which would involve analysing also indoor sources (smoking, frying, wood heating).

**Authors' response**

We totally agree that this is an important new research direction. Some of the present authors have studied this matter by detailed modelling for Helsinki (e.g., Kousa et al., 2002, Soares et al., 2014, Kukkonen et al., 2016) and for London (Smith et al., 2016, Vikas et al, 2019). These studies have allowed also for the movements of the population in various micro-environments, and the infiltration of pollution to indoor air. However, performing such modelling for an entire country would require a substantial amount of work. In our view, this could be a topic of another separate study.

**Author's changes in manuscript**

We have added a paragraph on dynamic exposure modelling to the conclusions section (next to the last paragraph).

**Referee's comment**

Another field is source apportionment to improve the large uncertainty of atmospheric modelling. In the text, a small chapter should at least mention current work on improving the methodology. A new part is the calculation of health impacts for emissions of different source groups like transport, residential a.s.o. Up to now, monetized health impacts have been estimated differentiated according to height of release and urban/nonurban area (see e.g. http://ecoweb.ier.uni-stuttgart.de/EcoSenseLE/current/index.php ). Some important (now older, but still valid) guidelines for the methodology could be cited, e.g. the ExternE Externalities of Energy Methodology 2005 Update; downloadable at http://www.externe.info/externe_d7/?q=node/30 or the IEHIAS Integrated Environmental Health Impact Assessment System (2011); web based guidance system; accessible under www.integrated-assessment.eu .

**Authors' response**

Emission and atmospheric modelling in this study has been substantially improved, compared with previous studies, especially concerning the national very fine resolution emission inventory and numerous recent refinements of the SILAM model. Some recent references have been added regarding the SILAM, FRES and UDM-FMI models, which illustrate the most recent advancements of these models.

We agree with the reviewer that a discussion and references on the EcoSenseLE and ExternE need to be added to the literature review and to the relevant sections in the text (e.g., Bickel and Friedrich, 2005). Clearly, these are very important research methods and activities in this area, and certainly need to be discussed and referenced.

**Author's changes in manuscript**

We have added discussion on the methodology of Bickel and Friedrich (2005), and the EcoSenseLE tool to the introduction (4th paragraph).

We have updated the references to the FRES model to include two latest published articles (section 2.1, first paragraph). The references regarding the SILAM model and its applications have been updated in section 2.2., first paragraph and section 2.2.1, first paragraph. One recent reference has been added on the evaluation of the UDM-FMI model (section 2.2.2).

We have also added to the conclusions section a discussion on model improvements in the present study, as per reviewer request, especially regarding the spatial resolutions and organic particulate matter. We have mentioned that one important chemical constituent of the $PM_{2.5}$ that has been neglected, according to our knowledge, by all of the previous published health cost studies has been here taken into account, viz. the organic particulate matter. We could take this fraction realistically into account, due to recent developments of the SILAM model. However, we felt that a more deep and thorough discussion on the improvement of chemical transport and other dispersion models would be outside the scope of this study.

**Referee's comment**

The result is indeed useful for consultants or decision makers in Finland, that want to identify the most efficient measures for reducing health effects from fine particles. They may use the given unit cost figures for estimating the health benefit caused by reduced emissions, and could also use the tool provided. The editors should decide, whether the publication fits into the scope of their journal or whether a journal more oriented towards environmental policy application might be a better choice

**Authors' response**

In the authors' view, the paper would fit best to the scope of ACP.

**Author's changes in manuscript**

No changes made.

**References**

Bickel, Peter and Rainer Friedrich (editors), 2005. ExternE Externalities of Energy Methodology 2005 Update. Institut für Energiewirtschaft und Rationelle Energieanwendung — IER Universität Stuttgart, Germany. Directorate-General for Research Sustainable Energy Systems, EUR 21951. Luxembourg: Office for Official Publications of the European Communities, 2004. ISBN 92-79-00423-9, European Communities, 2005, printed in Luxemburg, 270 pp.

Karvosenoja, Niko, Leena Kangas, Kaarle Kupiainen, Jaakko Kukkonen, Ari Karppinen, Mikhail Sofiev, Marko Tainio, Ville-Veikko Paunu, Pauliina Ahtoniemi, Jouni T. Tuomisto, Petri Porvari, 2010. Integrated modeling assessments of the population exposure in Finland to primary PM2.5 from traffic and domestic wood combustion on the resolutions of 1 and 10 km. Air Qual. Atmos. Health. DOI 10.1007/s11869-010-0100-9. http://www.springerlink.com/content/746378j3l307006v/fulltext.pdf.

Kousa, A., Kukkonen, J., Karppinen, A., Aarnio, P., Koskentalo, T., 2002. A model for evaluating the population exposure to ambient air pollution in an urban area. Atmos. Environ. 36, 2109–2119. https://doi.org/10.1016/S1352-2310(02)00228-5

Korhonen, Antti, Heli Lehtomäki, Isabell Rumrich, Niko Karvosenoja, Ville-Veikko Paunu, Kaarle Kupiainen, Mikhail Sofiev, Yuliia Palamarchuk, Jaakko Kukkonen, Leena Kangas, Ari Karppinen, Otto Hänninen. Influence of spatial resolution on population $PM_{2.5}$ exposure and health impacts. Air Quality, Atmosphere and Health (2019), Vol. 12, Issue 6, pp. 705-718. https://doi.org/10.1007/s11869-019-00690-z.

Kukkonen, J., Singh, V., Sokhi, R.S., Soares, J., Kousa, A., Matilainen, L., Kangas, L., Kauhaniemi, M., Riikonen, K., Jalkanen, J.-P., Rasila, T., Hänninen, O., Koskentalo, T., Aarnio, M., Hendriks, C., Karppinen, A., 2016. Assessment of Population Exposure to Particulate Matter for London and Helsinki, in: Steyn, D.G., Chaumerliac, N. (Eds.), Air Pollution Modeling and Its Application XXIV, Springer, pp. 99–105. https://doi.org/10.1007/978-3-319-24478-5_16

Soares, J., Kousa, A., Kukkonen, J., Matilainen, L., Kangas, L., Kauhaniemi, M., Riikonen, K., Jalkanen, J.-P., Rasila, T., Hänninen, O., Koskentalo, T., Aarnio, M., Hendriks, C., Karppinen, A., 2014. Refinement of a model for evaluating the population exposure in an urban area. Geosci. Model Dev. 7, 1855–1872. https://doi.org/10.5194/gmd-7-1855-2014

Singh, Vikas, Ranjeet S Sokhi and Jaakko Kukkonen, 2019. An approach to predict population exposure to ambient air $PM_{2.5}$ concentrations and its dependence on population activity for the megacity London. Environmental Pollution, *DOI:* 10.1016/j.envpol.2019.113623.

Smith, J.D., Mitsakou, C., Kitwiroon, N., Barratt, B.M., Walton, H.A., Taylor, J.G., Anderson, H.R., Kelly, F.J., Beevers, S.D., 2016. London Hybrid Exposure Model: Improving Human Exposure Estimates to NO2 and PM2.5 in an Urban Setting. Environ. Sci. Technol. 50, 11760–11768. https://doi.org/10.1021/acs.est.6b01817.

**Referee # 2 (Ståle Navrud)**

**Referee's comment**
This paper is an excellent and state-of-the-art application of the impact pathway approach (also termed the damage cost approach by economists) to assess public health costs of fine particulate matter (PM2.5) in ambient air from domestic primary and precursor emissions of PM2.5 in Finland in 2015. The paper provides a very transparent overview of the assumptions and calculations in all steps of the impact pathway model they develop; from emissions from different sources through dispersion and exposure to concentration-response functions and the economic valuation of the selected health endpoints.

**Authors' response**

Thank you for these positive comments.

**Author's changes in manuscript**

No changes made.

In the economic valuation of premature death due to PM2.5 the authors argue for using the number of lost life years and the Value of a Life Year (VOLY) rather than valuing all people at the Value of Statistical Life (VSL). This implies assigning a lower value to the elderly, that make up the majority of people dying prematurely due to PM2.5. The authors state in line 402-407: "The monetized estimates in the computations of the economic impacts in this study are based on the average value of a life year (VOLY), instead of the value of statistical life (VSL). The VOLY-based approach has been commonly used as a measure to assess a decrease in mortality risk (Im et al., 2018), whereas the VSL-based approach, despite its disadvantage, is in line with EPA's standard procedure and recommendations (Wolfe et al., 2019)". The "despite disadvantages "phrase should be deleted. They should instead say choosing between the VSL and VOLY approaches is also an ethical question in terms of whether one should assign the same economic value to all adults, independent of age; and that a constant VOLY assumes that people value a life year the same independent of age.

**Authors' response**

There was an unfortunate error in our text in the manuscript. In the method section 2.4. (lines 402-407 in the original manuscript), we should write that both VOLY and VSL -based methods were used. This is how we actually used the methods in the 'results' section, and how we described the conclusions.

We also agree that 'despite disadvantages' should be removed, and that selection between these two measures is partly an ethical question.

**Author's changes in manuscript**

We have corrected the error mentioned above (the revised manuscript version reads that we use both VOLY and VSL).

We have deleted the phrase "despite disadvantages", in relation to VSL. We have also somewhat restructured the 2nd and 3rd paragraphs of the section 2.4 to be more logical.

Also the authors further down in the paper (line 669-672) say that they have performed calculations using both VOLY and VSL (see also table 4), in order to show the difference in calculated economic benefits from emission reductions; which is of course very good that they do. However, in line 402 (see citation above) they state that they only used VOLY. This has to be corrected to made consistent with the calculations/sensitivity analyses they actually performed.

**Authors' response**

We thank the reviewer for finding this mistake in the original manuscript.

**Author's changes in manuscript**

We have corrected the mistake.

In line 399 after the sentence "This also facilitates numerical comparisons with those studies" they should add that: "Ideally a country-specific VSL (and a derived VOLY from this VSL) for Finland should be applied in this analysis, which could be based on value transfer (Navrud and Ready 2008) from the most recent global meta analysis of stated preference studies of VSL (Lindhjem et al 2011). However, this would preclude the direct comparison of results with similar impact pathway models e.g. Holland et al (2005).

**Authors' response**

We agree with the reviewer on this comment and thank him for providing this text.

**Author's changes in manuscript**

We have added this comment to the manuscript (2$^{nd}$ paragraph of section 2.4).

In the appendix the authors use VOLY as the abbreviation for Value of a Life Year , whereas in the text also VLY is used (e.g. in the paragraphs including lines 670, 685 and 705). This should be consistent, and I suggest using VOLY all the way through the paper.

**Authors' response**

We agree.

**Author's changes in manuscript**

All such acronyms have been corrected to read 'VOLY'.

They find that the economic benefits om emission reductions are clearly largest for the emission reductions for the source categories that have low emission heights, and are located in more densely populated regions (i.e. vehicular traffic, non-road and machinery and residential wood combustion urban versus rural areas). In the Abstract they say "It was found that economically the most effective measures would be the reduction of the emissions in urban areas of (i) road transport, (ii) non-road vehicles and machinery, and (iii) residential wood combustion". However, in economics cost-effectiveness, that this can statement can interpreted as, is a measure of e.g. reduced number of kg of PM 2.5 emissions per euro of abatement costs. Thus, the authors should rather say that " the economic benefits in terms of avoided public health costs is largest for measures that will reduce of (i) road transport, (ii) non-road vehicles and machinery, and (iii) residential wood combustion". Whether these measures have the largest net benefits (Economic health benefits minus the abatement costs) depends of course on the costs of reducing emissions from theses sources.

**Authors' response**

We totally agree with the reviewer, this is an important point.

**Author's changes in manuscript**

We have corrected this mistake in the abstract.

This also has to be corrected when using the word "effective" both in the abstract and in line 399-805.

**Authors' response**

We have checked the whole manuscript for the correct usage of terminology in this respect.

**Author's changes in manuscript**

We have revised the text at several instances, to be in accordance with the correct terminology, as per reviewer request.

There could also be other economic benefits than to public health from reducing the PM2.5 emissions, such as reduced soiling and corrosion of residential and commercial buildings, historic monuments and cultural heritage buildings as well as reduced environmental damages. The authors very briefly mentions this at the end of the paper, but they could also mention that there are ways to value these economic benefits and refer to e.g. Navrud and Ready (2002, 2007) and Watt et al (2009).

**Authors' response**

We agree on this comment.

**Author's changes in manuscript**

The first paragraph in the introduction has been extended to include these comments and references.

References:

Holland, M, A. Hunt, F. Hurley, S. Navrud, and P.Watkiss 2005: Methodology for Cost-Benefit Analysis of CAFÉ (Clean Air for Europe). Volume 1: Overview and methodology. Report to DG Environment. European Commission, 112 pp.

Lindhjem, H., S. Navrud, N.A. Braathen, and V. Biausque 2011: Valuing lives saved from environment, transport and health policies. A meta analysis. Risk Analysis 31 (9); 1381-1407.

Navrud, S. and R. Ready (eds.) 2002: Valuing Cultural Heritage. Applying environmental valuation techniques to historical buildings, monuments and artifacts. Edward Elgar Publishing.

Navrud, S and R. Ready (eds.) 2007: Environmental Value Transfer: Issues and Methods. Springer, Dordrect, The Netherlands. Watt, J., S. Navrud, Z. Slizkova and T. Yates 2009: Economic Evaluation.. Chapter 7 (pp. 189-214) In Watt, J., J. Tidblad, V. Kucera and R. Hamilton (eds.) 2009: The Effects of Air Pollution on Cultural Heritage. Springer. 306 pp.

**Author's changes in manuscript**

The above mentioned references were added to the reference list.

**Other minor revisions**

**Author's changes in manuscript**

In the affiliations on the first page, we have added the acronym 'SYKE', as this institutes prefers to write its name in this manner. The name of the third institute has also been updated. A second affiliation has been added to Prof. Lanki.